# RadWise: A Rank-Based Hybrid Feature Weighting and Selection Method for Proteomic Categorization of Chemoirradiation in Patients with Glioblastoma

**DOI:** 10.3390/cancers15102672

**Published:** 2023-05-09

**Authors:** Erdal Tasci, Sarisha Jagasia, Ying Zhuge, Mary Sproull, Theresa Cooley Zgela, Megan Mackey, Kevin Camphausen, Andra Valentina Krauze

**Affiliations:** Radiation Oncology Branch, Center for Cancer Research, National Cancer Institute, National Institutes of Health, Building 10, Bethesda, MD 20892, USA; erdal.tasci@nih.gov (E.T.); sarisha.jagasia@nih.gov (S.J.); zhugey@mail.nih.gov (Y.Z.); sproullm@mail.nih.gov (M.S.); theresa.cooleyzgela@nih.gov (T.C.Z.); mmackey@mail.nih.gov (M.M.); camphauk@mail.nih.gov (K.C.)

**Keywords:** glioblastoma, radiation, proteomic, feature selection, machine learning, pattern recognition

## Abstract

**Simple Summary:**

Glioblastoma (GBM) is a type of primary brain cancer that is extremely aggressive and almost always fatal. To examine the response to treatment and classify GBM non-invasively, researchers are turning to proteome analysis to identify protein biomarkers associated with interventions. However, interpreting large proteomic panels can be challenging, requiring computational analysis to identify trends. In this study, we aimed to select the most informative proteomic features that define proteomic alteration resulting from concurrent chemoirradiation (CRT) treatment which is standard of care for GBM following maximal surgical resection. We developed a novel rank-based feature weighting method called RadWise, which uses two popular feature selection methods, the least absolute shrinkage and selection operator (LASSO) and the minimum redundancy maximum relevance (mRMR), to identify relevant proteomic parameters. The computational analysis showed that RadWise outperformed other methods that did not employ a feature selection process, achieving high accuracy rates with very few selected proteomic features.

**Abstract:**

Glioblastomas (GBM) are rapidly growing, aggressive, nearly uniformly fatal, and the most common primary type of brain cancer. They exhibit significant heterogeneity and resistance to treatment, limiting the ability to analyze dynamic biological behavior that drives response and resistance, which are central to advancing outcomes in glioblastoma. Analysis of the proteome aimed at signal change over time provides a potential opportunity for non-invasive classification and examination of the response to treatment by identifying protein biomarkers associated with interventions. However, data acquired using large proteomic panels must be more intuitively interpretable, requiring computational analysis to identify trends. Machine learning is increasingly employed, however, it requires feature selection which has a critical and considerable effect on machine learning problems when applied to large-scale data to reduce the number of parameters, improve generalization, and find essential predictors. In this study, using 7k proteomic data generated from the analysis of serum obtained from 82 patients with GBM pre- and post-completion of concurrent chemoirradiation (CRT), we aimed to select the most discriminative proteomic features that define proteomic alteration that is the result of administering CRT. Thus, we present a novel rank-based feature weighting method (RadWise) to identify relevant proteomic parameters using two popular feature selection methods, least absolute shrinkage and selection operator (LASSO) and the minimum redundancy maximum relevance (mRMR). The computational results show that the proposed method yields outstanding results with very few selected proteomic features, with higher accuracy rate performance than methods that do not employ a feature selection process. While the computational method identified several proteomic signals identical to the clinical intuitive (heuristic approach), several heuristically identified proteomic signals were not selected while other novel proteomic biomarkers not selected with the heuristic approach that carry biological prognostic relevance in GBM only emerged with the novel method. The computational results show that the proposed method yields promising results, reducing 7k proteomic data to 8 selected proteomic features with a performance value of 96.364%, comparing favorably with techniques that do not employ feature selection.

## 1. Introduction

Gliomas are rapidly neurologically devastating, progressive, and the most common primary brain tumor originating from supporting cells of the brain called glial cells. Glioblastoma multiforme (GBM) is both the most malignant (WHO grade IV) and most frequently occurring subtype of CNS tumor [1,2], responsible for more than 60% of all brain tumors in adults [2,3]. The current standard of care (SOC) for GBM requires maximal surgical resection followed by concurrent chemoirradiation (CRT) comprising radiation therapy (RT) and temozolomide (TMZ), followed by adjuvant TMZ. However, this SOC management results in overall survival (OS) of less than 30% at two years [4,5]. Thus, GBM is still a deadly disease with an abysmal prognosis. 

The standard approach to GBM tumor classification is characterization using morphologic appearance and immunohistochemistry with the addition of imaging alterations using MRI [6]. In recent years, molecular biomarkers have gained importance within the World Health Organization (WHO) classification of central nervous system tumors [6].

The development of objective and robust diagnostic, prognostic, predictive, and treatment-assessing markers for deadly tumors has a critical priority and requirement to increase patients’ survival rates and facilitate oncologists’ decision-making process [7]. Biomarkers can also provide unique information related to response to therapy, and some may be disease mediators that need to be targeted for treatment [7]. The proteomic analysis yields rich information content to researchers at all levels, bench to bedside, by identifying proteins systematically at a specific time. Exploration and following alteration patterns in proteomic parameters for related diseases give a unique opportunity to diagnose, define prognosis, predict, and monitor the presence and progression of a specific pathologic process [7]. For example, plasma and serum have been excellent sample sources for human glioma biomarker research [7,8] in the last decade. 

Significant efforts are focused on methods for the early detection of tumors based on specific proteins or proteomic profiles generated from biopsies as well as serum and plasma [9,10]. The proteome provides a less invasive approach as compared to approaches that involve the need for tumor tissue or cerebrospinal fluid and is potentially informative to assess both the origin and progression of brain tumors to guide management [7].

There is currently no proteomic machine learning (ML)-based categorization study for pre- vs. post-completion chemoradiation therapy (CRT) in patients with GBM. Defining methods that explore proteomic alteration with treatment pre- vs. post-CRT, which is the pillar of SOC management in GBM, could generate proteomic signatures that can then be layered with clinical risk stratification to allow for interpretability downstream when linked to outcomes to identify potential proteomic biomarkers. 

In this study we focused on homing in on the method of curating proteomic signal alteration for future interpretability with clinical data. Pattern recognition is needed for this task. Feature selection employed for dimensionality reduction is one of the crucial stages in pattern recognition. Feature selection (FS) can be described as the selection of relevant features with the removal of redundant features to improve the model’s generalization performance. FS decreases model complexity by reducing the number of parameters, increasing learning speed, and circumvents the curse of dimensionality while enhancing data visualization [11,12,13,14,15]. The FS process can be employed in various medical scenarios, including medical imaging, biomedical signal processing, and DNA microarray data [15]. There are multiple studies illustrating its application to GBM data. To improve GBM prognosis prediction accuracy, [16] employed the minimum redundancy maximum relevance FS method (mRMR) using The Cancer Genome Atlas (TCGA) Program biomolecular features. For survival prediction in GBM, the dimensionality reduction stage was performed by Spearman’s correlation coefficient [17]. To differentiate GBM from brain metastases, the least absolute shrinkage and selection operator (LASSO) method was applied for the radiomic features to select a suitable variable subset [18].

Feature selection methods are commonly categorized into filter, wrapper, and embedded [11,19]. Filter methods are independent of the chosen learning model, selecting variables by employing various statistical measures such as correlation, statistical tests, and mutual information [20]. Wrapper methods evaluate the features for the performance of the used model and selection measure [20]. Embedded FS methods such as LASSO [21] carry out this FS process as an integral part of the learning process, often specific to a given learning machine [11,19]. Although FS methods yield more accurate results, the main drawback of FS is that the feature subset search space grows exponentially, requiring considerable computational time given that the number of predictors increases. For this reason, identifying fewer features with a high accuracy rate for high-dimensional data is a critical and challenging task in several application domains, including medicine. 

To this end, we introduce a novel, effective rank-based feature weighting (FW) and selection method to reduce the number of proteomic features and select the most relevant small subset. We apply this method to a large-scale proteomic oncologic dataset acquired before and after treatment to categorize its alteration, using two popular feature selection methods and various machine learning models. In this study, we examined CRT administration in patients with GBM. We focused not only on achieving high-performance results for the CRT classification tasks but also on achieving outstanding dimensionality reduction.

The main contributions of our study to the related literature are summarized as follows:To the best of our knowledge, this is the first study that utilizes a proteomic dataset acquired pre- and post-completion of CRT to categorize its alteration based on thousands of proteomic features available before and following intervention.To our knowledge, this is also the first study that employs filter and embedding-based FS algorithms for GBM proteomic data acquired with SOC treatment.We present a novel rank-based feature weighting and selection mechanism (RadWise) to identify relevant feature subsets with a cross-validation technique for machine learning problems.We combine the advantages of the two efficient and popular FS methods, namely, LASSO and mRMR, with rank-based feature weighting for pattern classification.We have investigated the comprehensive effects of FS and feature weighting methods separately for five different learning models on the proteomic dataset.The effects and the results of conventional statistical approaches without feature engineering have also been presented, compared, and discussed in detail and were compared to the FS method.We compared our proposed methodology with our statistical threshold-based heuristic method as well.We achieve high-performance results with approximately a thousand times smaller size features than the original proteomic dataset predictors.We examined the identified features in ingenuity pathway analysis (IPA) to link to disease (brain cancer, malignancy), pathways, and upstream proteins.Our results present promising results for GBM proteomic biomarker research in our field.

The following is the structure of the remaining sections of this study: First, we provide a summary of the suggested approach in Section 2, then we describe the applied feature selection and feature weighting methods with the associated feature selection techniques and supervised learning algorithms for CRT in patients with GBM. In Section 3, we explain the experimental process, describe the dataset employed and provide performance metrics with comprehensive experimental results. Next, we frame the proposed FS methodology and discuss our results in Section 4. Finally, Section 5 presents this study’s conclusion and possible future directions.

## 2. Methods

A brief overview of the proposed FS and weighting architecture for CRT status categorization (i.e., pre vs. post) of patients with GBM is presented in this section. We describe the feature selection, feature weighting, and classification methods employed for this study in the following subsections.

### 2.1. The Proposed Architecture for Feature Selection of Radiation Therapy Categorization of Glioblastoma Patients

Here, we describe a novel rank-based feature weighting and selection method (RadWise) to categorize the proteomic response to CRT in patients with GBM. Our methodology contains two main phases: (i) FS and (ii) feature weighting. Figure 1 outlines a brief overview of the proposed architecture and the related processes with an example drawing. Each feature is illustrated in a different color, with features selected to move forward represented by an x sign, with the other features removed from the set in the FS step. In the feature weighting phase, feature set lists are constructed according to weight values by employing FS methods with a cross-validation technique.

Initially, all proteomic values with human or non-human characteristics are filtered to select only human-based proteomic features, followed by the designation of the sample (acquired pre- or post-CRT) being directed into the FS phase using a cross-validation technique. Every selected feature set identified by the two different feature selection methods utilized for each fold is saved, and their counts are increased by using the corresponding FS method weights (i.e., rank-based approach). Subsequently, the minimum weight-based feature list is evaluated with all of the different weight values. In the last stage, the final selected feature list is obtained by assessing all the weight values to identify those with the highest accuracy rate.

The detailed algorithmic view of the proposed architecture is illustrated in Figure 2, which shows the two FS methods, LASSO and mRMR. These methods choose features for every fold of the cross-validation of the human features-based proteomic dataset. Then, the weights of the selected features in each set for each fold are increased according to the rank-based importance level of the chosen FS methods (i.e., LASSO features are increased by one, and mRMR features are increased by two according to their performance results in terms of accuracy rate, see Table 1). For example, if the same feature is selected by both LASSO and mRMR FS methods for each fold of 5-fold cross-validation, the weight of this feature will be 15 (i.e., the summing of assigning 2 as a weight value with mRMR and of 1 as a weight value with LASSO for 5-fold cross-validation). This value is the maximum weight value for 5-fold cross-validation. We obtain a 0 value for the minimum weight value of the selected feature in a similar manner if the feature is not chosen by any FS method for each iteration. However, we should identify this value at least as 1 to use all selected features of all iterations of cross-validation for the experimental results. 

After the FS and feature weighting stages, total weights with the corresponding feature lists are obtained, and the evaluation of minimum weight-based feature lists and selection of final features are performed for all weight values. To better explain this process, we can give an example. If we identify the minimum weight as 13 for our study, we can use the selected features with the weight values of 13, 14, and 15 (e.g., [F1, F3, F4, F6, F8, F11, F14]) as the final selected feature set according to Figure 1. These weight values are evaluated according to their performance results in terms of accuracy rate. We identify the min weight value with the highest accuracy rate and the minimum selected number of features for all values set in the dataset in this study. 

### 2.2. Feature Selection Methods

Feature selection methods obtain a suitable feature subset from all feature sets by reducing the dimensionality of data space. Feature selection methods aim to eliminate redundant, insignificant, or irrelevant features, reduce the time requirement of the training phase of the learning model, accelerate prediction speed, and yield better model interpretation and diagnosis capability as a data preprocessing step [22,23,24,25]. In real-world applications, FS methods also deal with dimensionality problems (e.g., high-dimensional data), computational and storage complexity, data visualization, and high-performance issues for machine learning-related problems [26].

Feature selection (FS) methods are broadly divided into three categories depending on the evaluation metric of the feature subset of the process as a filter, wrapper, or embedded method. Filter methods are also categorized into univariate filter or multivariate filter FS methods depending on considering the relationships between the features and/or between the features and target/class or output variable. In this study, we have utilized an ensemble of a popular and highly efficient multivariate filter FS method, the minimum redundancy maximum relevance (mRMR), and an efficient embedded-based FS method, LASSO, for the selection of the proteomic features for GBM patients’ data. Given that the wrapper FS method is more computationally intensive and to avoid the risk of dependence on model-specific features that may be associated with this approach, we chose to take advantage of the filter and embedded FS methods for this study to enhance the transferability of the approach. We describe the two employed FS methods (i.e., mRMR and LASSO) for this study in the following subsections. 

#### 2.2.1. mRMR

mRMR is a multivariate filter FS method that employs heuristic techniques, introduced by Peng et al. in 2005 [27]. This method can be used for continuous and discrete high-dimensional datasets [27,28]. This approach selects features assessing both the relevance for predicting the target variable and the redundancy within the selected predictors [25]. The main purpose of mRMR is to find the best feature collection that jointly has the largest dependency or relevance on the output class or has minimal redundancy on the selected feature subset [28].

mRMR assumes that there are in total m features and, for a given feature Xi (I ∈ {1, 2, 3, …, m}), its feature importance based on the mRMR metric can be defined as Equation (1) [25,27].

(1)
$$f^{\mathrm{mRMR}}\left(X_{i}\right)=I\left({Y,X}_{i}\right)-\frac{1}{\left|S\right|}\underset{X_{s}\in S}{\sum }I\left(\mathrm{Xs},\mathrm{Xi}\right)$$

where Y is the output/class label variable, S is the set of selected features, |S| is the size of the feature set (number of features), Xs ∈ S is one feature out of the feature set S, Xi shows a feature currently not selected: Xi ∈ S.

For discrete variables X and Y, the mutual information formula is in the form of Equation (2) [27]:
(2)
$$I\left(Y,X\right)=\underset{y\in \omega_{Y}}{\sum }\underset{x\in \omega_{X}}{\sum }p\left(x,y\right)\log\left(\frac{p\left(x,y\right)}{p\left(x\right)p\left(y\right)}\right)$$

where 
$\omega_{Y},\omega_{X}$
 are the sample spaces related to *Y* and *X*, *p*(*x*, *y*) is the joint probability density, and *p*( ) is the marginal density function.

In the mRMR feature selection process, at each iteration, the feature with the highest feature importance value 
$max_{X_{s}\in S}$

$f^{mRMR}\left(X_{i}\right)$
 is added to the selected feature set *S* [25].

#### 2.2.2. LASSO

LASSO, the abbreviation of least absolute shrinkage and selection operator, is a statistical formula and linear model used for reducing the dimension of the problem, regularization, and regression analysis in statistics and machine learning [12,29,30]. This operator estimates sparse coefficients of parameters. The LASSO method consists of a linear model with an added regularization term. Its goal is to minimize the objective function illustrated in Equation (3) [31]. Assume y = (y_1_, …, y_n_)^T^ is the response vector. Additionally, x_j_ = (x_1j_, …, x_nj_ )^T^, j = 1, …, p are the linearly independent predictors [32]. Let predictor matrix be X = [x_1_, …, x_p_]. Consider the data values to be standardized. The coefficients of a linear model estimated by LASSO are provided by Equation (3) [32].

(3)
$$\hat{\beta }=\underset{\beta }{arg\;min}\|y-\overset{p}{\underset{j=1}{\sum }}x_{j}\beta_{j}{\|}^{2}+\lambda \overset{p}{\underset{j=1}{\sum }}\left|\beta_{j}\right|$$


The LASSO regularization parameter *λ* and its exact unbiased estimate of the degrees of freedom 
$\hat{\beta }$
 can be used to create an adaptive model selection metric for efficiently selecting the optimal LASSO fit [12,32]. As a special case of the penalized least squares regression with an L1-penalty function, the LASSO method can be beneficial in terms of both model accuracy and interpretability, as it can remove irrelevant or insignificant features from the pattern, especially when there is a high correlation in the feature groups, as it will choose only one among them and shrink the coefficients of others to zero [31]. 

### 2.3. Feature Weighting

In the feature weighting phase, each feature is generally added or multiplied by a weight value proportional to the importance level of the feature to discriminate pattern classes for the feature selection operation [33]. The rank-based feature weighting approach has been applied to this study. In this approach, we ranked the two FS methods (LASSO and mRMR) based on their performance results (i.e., accuracy rate). Then, we assigned a higher rank/weight to FS method. Given the high efficiency and more accurate results obtained using the mRMR FS method on the high-dimensional dataset as compared to the LASSO method, we assigned the weight value of 2 to the related features in the selected feature list of each fold, adding this to the total weight list of features for the mRMR. These operations were repeated, assigning a weight value of 1 to the LASSO FS method (see Table 1 for performance comparison).

### 2.4. Classification Methods

In pattern classification, classification is a supervised learning technique that assigns a class label to a given set of data points based on their characteristics. Supervised learning is the most crucial methodology in machine learning and requires an algorithm mapping between a set of input features X and an output label Y and applying this mapping to predict the outputs to classify other unlabeled data [34,35]. The supervised learning and classification stage is important for solving complex and computational real-world problems.

In this study, five supervised learning algorithms (support vector machine, logistic regression, random forest, K nearest neighbors, and AdaBoost) were used for machine learning tasks. In the next subsections, we briefly describe these learning models used in the experimental study.

#### 2.4.1. Support Vector Machine

Support vector machine is one of the most well-known supervised learning algorithms, introduced by Vapnik et al. [36,37] for classification tasks. Support vector machines are a type of machine learning algorithm that uses structural risk minimization and statistical learning theory [38] to classify binary data points. Aizerman et al. [39,40] proposed various kernel tricks to map the input data points to a higher dimensional space, and the SVM model attempts to find a hyperplane with the greatest margin to separate the two classes. This algorithm is practical and effective for binary and linear or non-linear classification problems [12,38].

#### 2.4.2. Logistic Regression

Logistic regression, also known as the logit model, is a widely used and straightforward statistical tool for predicting and categorizing data. Binary logistic regression is often used for modeling the relationship between a categorical outcome and a set of predictors [41]. The logistic regression model assigns each feature a coefficient that calculates how much each feature contributes to the variation in the dependent variable [12,42]. 

#### 2.4.3. K Nearest Neighbors

K nearest neighbors model is one of the most straightforward classifiers in machine learning techniques [43]. This algorithm is a simple, effective, easy-to-understand, widely used, basic classification model and lazy learning algorithm [44,45]. The K nearest neighbors model categorizes unlabeled or test data by assigning them to the class of the most similar labeled examples with respect to distance function and the number of neighbors metric. 

#### 2.4.4. Random Forest

Random forest, proposed by Leo Breiman [46], is one of the most accurate, fast, simple, computationally effective, and easily parallelized general-purpose classification models and most robust to outliers and overfitting [47,48]. Random forest constructs an ensemble learning model with many combinations of decision tree predictors that grow in randomly selected data subspaces by adding some randomness to the bagging approach [47]. Different and many decision trees are used to categorize new data by the majority vote. Each decision tree model uses a feature subset randomly chosen from the original set of features [49]. Each tree uses different bootstrap sample data, similar to the bagging method [49].

#### 2.4.5. AdaBoost

AdaBoost, the abbreviation of adaptive boosting, is a boosting method and ensemble learning technique in machine learning [12]. Boosting algorithms employ many weak learners to construct a robust model. AdaBoost relies on observations that were previously incorrectly classified, and, in each iteration, the weights of misclassified samples are increased to be paid more attention [50], while the weights of correctly classified samples are decreased [51]. These operations are repeated until the optimal results are obtained. After iterations, the output of the base learners is combined into a weighted sum that forms the final output of the predictive model [50].

## 3. Experimental Work

This section describes the experimental processes and environment, relevant parameters, and performance metrics and explains our constructed and employed proteomic dataset. Then, we present our comprehensive computational results in conjunction with observing the effects of different feature selection methods in detail. Finally, we compare our results with respect to using only statistical approaches for proteomic biomarker identification on the proteomic dataset.

### 3.1. Experimental Process

In this study, to apply the proposed methods, we utilized the Python programming language with the scikit-learn [52] package for machine learning-related tasks and the mRMR [53] package for a filter-based feature selection process. We conducted all the experiments on a MacBook Pro notebook system with macOS Ventura, Intel Core i9 CPU with 2.3 GHz 8-core, and 16 GB 2667 MHz DDR4 RAM. There are five employed predictive models, including SVM, LR, KNN, RF, and AdaBoost, for the feature selection and classification tasks. We assigned default values as the corresponding parameter values to the related classifiers (i.e., C = 1, kernel = ‘rbf’, gamma = ’scale’, class_weight = ‘None’ for support vector machine; num of neighbors = 5, metric = ‘minkowski’, weights = ‘uniform’ for K nearest neighbors; penalty = ‘l2’, dual = False, tol = 0.0001, C = 1.0, fit_intercept = True, class_weight = None, intercept_scaling = 1 for logistic regression; n_estimators = 100, criterion = ‘gini’, min_samples_split = 2, max_depth = None, min_samples_leaf = 1, min_weight_fraction_leaf = 0.0 for the random forest; n_estimators = 50, algorithm = ‘SAMME.R’, learning_rate = 1.0 for AdaBoost). We adjusted the random state number to 0 for all five learning models to obtain the same value with the same random state on the proteomic datasets used.

To mitigate bias in the selection process of the proteomic features, we received and used the normalized proteomic data values for our empirical analyses from an aptamer-based SomaScan™ proteomics assay technology of SomaLogic [54]. In addition to the normalized data values, since we have a small number of cases, we applied a five-fold cross-validation technique for the feature selection and weighting processes and also obtained mean performance results of the learning models utilized. 

In this study, the post-CRT and pre-CRT classes have been considered positive and negative, respectively, for assessing statistically learned models. Both mRMR and LASSO methods have been applied for the hybrid feature selection stage. For the mRMR-based FS, we used the logarithmic value of the number of all proteomic features as a heuristic value (i.e., ⌈log_2_(Total Number of Features)⌉ = round of log_2_(Total Number of Features(7289)) = 13) to identify the number of features selected. For the LASSO-based FS operation, 10-fold cross-validation has been performed to acquire the best alpha parameter value utilizing the iterations and in order to find the number of features selected.

For the feature weighting stage, a rank-based approach in proportion to the feature selection methods’ performance results (i.e., accuracy rate) has been adopted. To identify the final selected feature set, all possible minimum weight values (i.e., from 15 to 1) have been tried to find the relevant subset with the best accuracy rate and the minimum number of features. 

### 3.2. Dataset

In this study, 82 patients with pathology-proven GBM (diagnosed 2005–2013) enrolled on NCI NIH IRB approved protocols and treated with CRT, with serum biospecimens obtained before and upon completion of CRT (i.e., the total number of cases is 164), were included. The proteomic dataset storing operations were provided by NIDAP [55]. 

Serum samples were obtained before CRT (average seven days, range 0 to 23) and following its completion (average seven days, range −1 to 30), with the time between pre- and post-sample acquisition averaging 49 days (range 27–83 days). Following the acquisition, the samples were frozen at −80° for an average of 3442 days (range 800–5788 days or 2.2–15.9 years) and then defrosted and screened using the aptamer-based SomaScan™ proteomic assay [54] technology from SomaLogic for changes in expression of 7596 protein analytes [56,57]. SomaScan™ data were filtered to remove non-human and non-protein targets, resulting in 7289 aptamers (i.e., proteomic features).

### 3.3. Performance Metrics

To evaluate the FS performance of the proposed methodology, we have employed six different evaluation metrics including classification accuracy (ACC), F-measure (F1), area under the ROC curve (AUC), precision (PRE), recall (REC), and specificity (SPEC) [58]. 

The classification accuracy rate is calculated by dividing the total number of true negatives and true positives by the total number of false positives, true positives, false negatives, and true negatives. The corresponding equation is shown in Equation (4).

(4)
$$\mathrm{ACC}=\frac{\mathrm{TP}+\mathrm{TN}}{\mathrm{TP}+\mathrm{TN}+\mathrm{FP}+\mathrm{FN}}$$

where TP, TN, FP, and FN denote the number of true positives, true negatives, false positives, and false negatives, respectively.

AUC, the two-dimensional area under the receiver operating characteristic (ROC) curve, is a commonly used performance metric constructed by plotting the true positive rate against the false positive rate for the classification model. A high AUC score shows that the classifier will perform well. AUC range values can be between 0 and 1.

F-measure (i.e., F1-score) is calculated by the harmonic mean of precision and recall. A perfect classification algorithm has an F-score of 1. It is defined in Equation (5).

(5)
$$F1=\frac{2\;*\;\mathrm{PRE}\;*\;\mathrm{REC}}{\mathrm{PRE}+\mathrm{REC}}$$


Precision means the positive predictive value. It is the fraction of the number of true positives to the total number of true positives plus false positives. The related equation is shown in Equation (6).

(6)
$$\mathrm{PRE}=\frac{\mathrm{TP}}{\mathrm{TP}+\mathrm{FP}}$$


Recall, known as sensitivity, is the true positive rate/hit rate or sensitivity. Recall is defined as division of the number of true positives by the total number of true positives plus false negatives. The equation is illustrated in Equation (7).

(7)
$$\mathrm{REC}=\frac{\mathrm{TP}}{\mathrm{TP}+\mathrm{FN}}$$


Specificity is defined as the true negative rate. Specificity is calculated by dividing the number of true negatives by the total number of true negatives plus false positives. The equation of specificity is given in Equation (8).

(8)
$$\mathrm{SPEC}=\frac{\mathrm{TN}}{\mathrm{TN}+\mathrm{FP}}$$


### 3.4. Computational Results

In this subsection, we describe the experimental results to illustrate the various effects of the FS and weighting approaches on the performance of the models in this study. Bold values indicate the most optimal results. In subsequent tables color changes from red to green display performance results from the lowest (red) to the highest values (green). Bold values indicate the best result for each method.

#### 3.4.1. The Effects of Using Feature Selection Methods

In the first phase, we conducted experiments using five supervised learning models to observe whether there is any benefit of FS procedures on the proteomic dataset employed. The mean performance results of the five supervised models in terms of accuracy rate (%) are illustrated in Table 1 and highlighted in Figure 3 to show the effects of using all features (i.e., no FS) and applying FS procedures individually in this study. 

As shown in Table 1, the supervised learning model without FS resulted in the lowest accuracy rate values for each approach. When comparing the mRMR FS and LASSO FS methods, mRMR yielded superior results across all five supervised learning models. Additionally, the best accuracy rate value of 92.708 resulted from combining the mRMR FS method and LR model. Using LASSO FS, the best accuracy rate values of 89.072 with the AdaBoost model and, without FS, the best accuracy rate value of 88.409 were obtained with the AdaBoost model. After obtaining the results of this stage, we moved the FS process to the next level (i.e., feature weighting) by assigning corresponding ranks to these FS methods accordingly.

#### 3.4.2. The Effects of Using Only LASSO Feature Selection and Weighting Methods

In the second phase of the empirical analysis, we conducted corresponding experiments related to various FS and weighting-based approaches to identify the best methods for this study. We evaluated the performance results of five classification models using only LASSO FS and weighting methods, as illustrated in Table 2. The minimum weight value is related to finding the minimum number of features from all sets. We employed a 5-fold cross-validation technique due to the number of instances in this study. In this case, the minimum weight value can be five as the maximum value to find the minimum of the selected features when the same feature is chosen for every iteration of five-fold cross-validation. This is also similarly the case for the minimum value. The minimum weight value can be 1 as the minimum value to use at least one selected feature from all feature sets for 5-fold cross-validation. According to these results, the best result was obtained with an accuracy rate value of 96.363, a minimum weight value of 3, and a number of selected features of 44 using the LR classifier. The second-best result was obtained with an accuracy rate value of 93.939, a minimum weight value of 4, and a number of selected features of 26 using the RF. We aimed to find the best accuracy rate with the minimum number of selected features by trying both FS and feature weighting methods individually and in combination. This approach results in the best accuracy rate with a value of 96.363.

#### 3.4.3. The Effects of Using Only mRMR Feature Selection and Feature Weighting Methods

The computational results for mRMR-based FS and weighting methods in terms of accuracy rate (%) are presented in Table 3. The best result was obtained with an accuracy rate value of 96.364, a minimum weight value of 3, and a number of selected features of 8 using the logistic regression model. The second-best result was obtained with an accuracy rate value of 96.364, a minimum weight value of 2, and a number of selected features of 11 using the logistic regression model.

#### 3.4.4. The Effects of Using LASSO and mRMR Feature Selection and Feature Weighting Methods

The computational results achieved when employing both LASSO and mRMR-based FS with weighting methods are reported in Table 4 and Table 5 as mean ± standard deviation. The most optimal result obtained resulted in an accuracy rate value of 96.364, a minimum weight value of 10, and a number of selected features of 8 using the logistic regression model. The second-best result had an accuracy rate value of 96.364, a minimum weight value of 9, and a number of selected features of 8 using the logistic regression model. The number of selected features resulting from each of these was the same. Thus, the selection of the correct minimum weight had not impacted the results. If the number of features emerging had been different with the same accuracy rate given different weight values, the minimum weight value could have been set as the maximum value to allow the selection of the minimum number of features. The best result for this method also provided the best performance among all methods applied. The method allowed for the identification of the names of the proteomic features selecting 8 features from the 7k reported in the proteomic dataset. The names of the proteins are as follows: K2C5, MIC-1, GFAP, STRATIFIN, cystatin M, keratin-1, CSPG3 and PRTN3.

Figure 4 shows the effects of the number of features related to the minimum weight value using LASSO and mRMR-based feature selection with weighting methods (see Table 4). The number of selected features increases significantly when the minimum weight value is lower than four (e.g., 3, 2, or 1). We also show the minimum weight and classification model-based effects in terms of accuracy rate as a column chart graphic in Figure 5. We observed that the logistic regression model generally provided the best results on the proteomic dataset for this study.

#### 3.4.5. Performance Results Based on Feature Selection and Weighting Process 

We obtained comprehensive computational results based on feature selection and weighting process using six performance metrics: the accuracy rate (ACC) %, area under the ROC curve (AUC), F-measure, precision, recall, and specificity. The detailed performance results without FS and weighting are illustrated in Table 6. The AdaBoost model provided the highest values without FS and weighting process resulting in 88.409, 0.951, 0.886, 0.882, 0.893, and 0.873 for the ACC, AUC, F-measure, precision, recall, and specificity, respectively. Performance results post-LASSO and mRMR-based FS with weighting operation are presented in Table 7. The logistic regression model resulted in the highest values with LASSO and mRMR-based FS and weighting process with 96.364, 0.987, 0.964, 0.963, 0.965 and 0.965 for ACC, AUC, F-measure, precision, recall, and specificity, respectively. The SVM model provided the best AUC value of 0.989 in this process.

### 3.5. Comparison with the Related Methods for Proteomic Biomarker Identification

We compare the proposed rank-based feature weighting and selection method with related methods for proteomic biomarker identification on the proteomic dataset used in this subsection.

#### 3.5.1. Comparison with Statistical Threshold-Based Heuristic Method (OSTH) 

As an initial step, we conducted a preliminary analysis to evaluate if the signal was present in the data with a heuristically threshold-based approach to identify the most substantial components of the proteomic panel in this context. A possible approach to a novel large protein panel where the most critical signals have yet to be defined can include the selection of the most altered protein targets in the most significant number of patients to elicit the selection of the most significant target protein signal. This type of analysis fixes two thresholds for filtration of the protein signal: The potential strength of the signal and the number of instances (number of patients affected by the signal alteration). To frame the eventual ML approach described in this study, this initial heuristic approach was also employed. We noted that several protein targets identified in the heuristic approach are identical to those identified in the ML approach, specifically, the top 4 elevated protein targets: K2C5, keratin-1, MIC-1, and STRATIFIN (Appendix A). The main reason for this finding, despite not employing FS, is that the ML approach itself primarily employs a filtration approach, and we assigned a higher priority to the filter-based feature selection method called the minimum redundancy maximum relevance (mRMR), which has a higher rank value than the embedded-based feature selection method called LASSO in this study, given its superior performance in experiments carried out as part of the study. Thus, the mRMR method uses statistical measures (i.e., correlation) for the feature selection step. This method tends to select a subset of features having the most correlation with the class (target) and the least correlation between themselves. As a result, if we use statistical approaches such as selecting proteins with >50% upregulation/elevation values after CRT in the most significant possible number of patients, we should invariably arrive at several similar selected features given the dataset obeying the thresholds as mentioned earlier. However, this type of heuristic logic will allow for neither selection of minimum redundancy between the chosen features nor the identification of an exact unbiased estimate of the degrees of freedom as would be the case with the LASSO approach and is thus limited since the characteristics of the underlying model learning and training process are not considered. We observe that several protein targets that were heuristically selected are likely redundant since they were not chosen by any other approach, e.g., TSH, although they were altered in a relatively large proportion of patients. By contrast, several targets identified in the ML approach and the statistical test-based method (Table 7) would be missed when employing the heuristic method as they would not meet the abovementioned thresholds. The most significant shortcoming of the heuristic approach is the need for more ability to identify the exact threshold value that maximizes the number of parameters of significance in the absence of FS. An additional point that renders FS superior is adding, removing, or optimizing features/targets toward the final variable set using FS techniques. Heuristic approaches such as the one described here significantly limit the optimal selection of clinically actionable or essential endpoints. 

#### 3.5.2. Comparison with the Related Statistical Test-Based Method 

When employing a statistical test-based approach [60], all the proteins identified by ML are also identified in this statistical test-based analysis. However, we noted the selection of the cystatin M protein which did not carry the same level of significance indicated in the other signals but was selected in the ML approach and carries biological relevance to glioma (Table 8). We wanted to explore the interpretability of the ML approach in choosing this feature, and this can be summarized as follows:Our feature selection approach employs two strategies: Multivariate filter FS (i.e., mRMR) and embedded FS (i.e., LASSO).Our hybrid FS method utilizes cross-validation to obtain a more robust result and a final feature set for our dataset.Our rank-based weighting approach assigns more importance to mRMR than to LASSO to arrive at the described performance results (i.e., accuracy rate). Due to this weighting criterion, the most prominent features identified in the statistical analyses are also identified by ML. However, signals such as cystatin M that in the purely statistical test-based approach rank lower can be elevated given their interaction with other targets. This is of great importance since novel large proteomic panels may identify molecules with known, unknown, or unassigned biological annotation and eventually novel functions or roles that are context specific.We experimented with all minimum weight-based values to obtain the minimum selected features with the highest accuracy rate. This process reduced 7289 protein signals to 8 with a higher accuracy rate.If the current feature selection process were not applied, the number and names of the features could not be determined precisely.

**Table 8 cancers-15-02672-t008:** Overview of the identified proteomic biomarkers illustrating the biological relevance to glioma.

Entrez Gene Symbol	Target Full Name	Biological Relevance to Glioma
K2C5	Keratin, type II cytoskeletal 5	Yes, evolving biomarker/target [61]
Keratin-1	Keratin, type II cytoskeletal 1	Yes, evolving biomarker/target [61]
STRATIFIN (SFN)	14-3-3 protein sigma	Yes, tumor suppressor gene expression pattern correlates with glioma grade and prognosis [62]
MIC-1 (GDF15)	Growth/differentiation factor 15	Yes, biomarker, novel immune checkpoint [63]
GFAP	Glial fibrillary acidic protein	Yes, evolving biomarker/target [64]
CSPG3 (NCAN)	Neurocan core protein	Yes, glycoproteomic profiles of GBM subtypes, differential expression versus control tissue [65]
Cystatin M (CST6)	Cystatin M	Yes, cell type-specific expression in normal brain and epigenetic silencing in glioma [66]
Proteinase-3 (PRTN3)	Proteinase-3	Yes, evolving role, may relate to pyroptosis, oxidative stress and immune response [59]

#### 3.5.3. Connecting the Identified Markers to Disease, Pathways, and Upstream Proteins

To investigate the ML-identified markers towards elucidating clinical relevance, potential connections were analyzed two-fold using ingenuity pathway analysis (IPA) (QIAGEN Inc., https://www.qiagenbioinformatics.com/products/ingenuitypathway-analysis) [67] (accessed on 5 April 2023) by employing the signal alteration log change and *p*-values and analyzing their footprint in terms of disease and functions (Figure 6) (Appendix A) and their alteration in relationship to upstream mediators (Figure 7) (Appendix A). In doing so we noted that all 8 protein signals mapped primarily to the cancer and organismal injury and abnormalities category and neoplasia of cells while 5 of the 8 signals (GDF15, GFAP, KRT1, KRT5, NCAN) mapped to brain cancer among disease and function annotation. Two of the eight (KRT1 and KRT5) additionally mapped to glucorticoid receptor signaling (Figure 6A,B). The dominant pathways that emerged were 14-3-3 mediated signaling (associated with cell cycle regulation and apoptosis) and Pi3K/AKT signaling (associated with intracellular and second messenger signaling, cellular growth, proliferation and development and cancer) (Figure 6C,D). These relationships require further analysis and validation, however. Given the dimensionality of the proteomic signal and the need to determine how signal alteration connects to possible mediators of management and the malignant process, we employed IPA to explore the connection of the identified signals to upstream mediators to infer possible mechanistic relationships (Figure 7). EGF (Figure 7A) and CRNNB1 (Figure 7B) were the top 2 upstream proteins collectively associated with 6 of the 8 signals (GDF15, GFAP, KRT1, KRT5, NCAN, SFN). The interaction between the identified features that emerged using the IPA-generated merged network for the 8 ML-identified proteins using the disease classification brain cancer is illustrated in Figure 7C. 

## 4. Discussion

Proteomic data are growing with the increased acquisition of biospecimens and the growth of protein panels generating increasingly large data. Proteomic data are expensive to acquire and often originate from a relatively small number of patients, resulting in a small number of instances for analysis overall. However, proteomic data are highly multidimensional [68], requiring feature engineering to optimize their use. The application of machine learning towards proteomic data is evolving [69] with applications towards protein–protein interactions [70], response to intervention [71], and biomarker discovery [68,72]. The method we describe here is aimed at two interventions that occur concurrently, chemotherapy and radiation therapy in patients with a histologic diagnosis of GBM. Currently, no proteomic panels of this scale have been characterized for either radiation therapy or chemotherapy in GBM. Given that CRT is SOC, proteomic characterization of either intervention in isolation is not likely to be forthcoming. The goal of the method presented here was to define the alteration in the proteome in patients with GBM who undergo CRT in the most efficient manner possible, while also allowing for interpretable results that can be validated to advance the field. The maximum relevance minimum redundancy (mRMR) algorithm, least absolute shrinkage and selection operator (LASSO) algorithm, and the five supervised learning models consisting of logistic regression, support vector machine, K nearest neighbors, random forest, and AdaBoost classification algorithms have all been employed in oncology settings including nomograms [73], genome-wide methylation analysis [74], and prediction of outcomes after CRT [75], however, not in the context of the proteomic alteration. Several results and conclusions emerged based on the comprehensive experimental results of our novel hybrid feature selection and rank-based weighting method (RadWise). Our novel rank-based feature weighting procedure identified relevant feature subsets with a cross-validation technique for machine learning problems, which is an essential step in the study. Given that determining the most important features in the cross-validation process would result in different features for every fold of the cross-validation technique, it was imperative to provide a method that allowed for the identification of the names of the most significant features. In this case, these were protein targets, as they are clinically highly relevant to advance the inclusion of findings into basic science examination and validation in other datasets generated both at the bench and bedside. The method presented here uniquely provides the names of the selected features by assigning higher weights to the features of one FS method and lower weights to the features of another in each fold. The method also allows the setting of minimum weight values to finalize the feature subset in the study. Eight proteomic features were thus identified in this large proteomic panel that defined the administration of CRT in 82 patients with GBM based on biospecimens obtained prior to and after treatment. The identified proteomic features K2C5, MIC-1, GFAP, STRATIFIN, cystatin M, keratin-1, CSPG3 and PRTN3 add to the knowledge base as this is the only large-scale proteomic dataset derived from biospecimens captured prior to and after administration of treatment. This provides a unique window into how the proteome is altered with management and a unique opportunity to explore feature engineering to harness large omic data. The molecules identified with the novel method described here provided the best accuracy rate with the minimum number of selected features. All eight proteins are associated with biological relevance in glioma, one representing a highly studied tumor suppressor gene 14-3-3 [62], another a novel immune checkpoint [63], with four of eight representing evolving biomarkers [59,61,64], and two of the eight exhibiting differential expression versus normal tissue [65,66]. All eight were also identified in analyses we previously carried out without ML [60]. Most importantly, we observed that some molecules that may be identified heuristically could be eliminated using this method. In contrast, molecules that otherwise may not reach detection limits in statistical analyses, but carry significant biological relevance, may emerge while employing this approach. The ML method identified protein targets that, based on current literature and additionally exemplified via IPA, may exhibit differential expression versus normal tissue and connections to irradiation [76], and this merits further analysis in conjunction with clinical, radiation therapy dosimetry dose volume parameters, and patient imaging. These upstream connections may represent connections to the administration of CRT as well as tumor proliferation. Of particular interest, the association with glucocorticoid receptor signaling may relate to the co-administration of corticosteroids to mitigate inflammation and resulting neurological symptoms in patients with GBM, requiring further analysis in conjunction with medication administration and dosing attributes. While this set of molecules requires further validation, there is a demonstrated connection to cancer, brain cancer, and known pathways associated with both as well as the administration of CRT. These molecules can act as a template for CRT proteomic panels in patients with GBM which can then be leveraged against various RT dose scenarios and the addition of novel agents to SOC. Considering the predictive performance of the hybrid feature selection method employed for our framework in this study, our proposed FS method outperforms the results of using either LASSO or mRMR-based FS methods in isolation in terms of both accuracy rate and the number of selected features. The FS of 8 out of 7289 proteins provides approximately 1000 times performance gain regarding the number of features for our study. This allows researchers to focus on lower-dimensional data/feature space, which enhances decision-making or diagnostic processes in future studies from bench to bedside. The limitation of this study is that we focused on only two commonly employed FS methods to arrive at the relevant feature subset. The feature subset may vary, given alternative FS methods. However, the results arrived at here improve upon other literature results in this space and can act as a basis for further augmentation and experimentation for researchers.

## 5. Conclusions and Future Work

This study presents a novel hybrid FS and rank-based weighting method (RadWise) for categorizing proteomic response to CRT in glioblastoma patients using a large-scale proteomic dataset obtained prior to and after completion of treatment. LASSO and/or mRMR-based FS and weighting methods have been applied to identify the best feature subset that defines this intervention based on the proteome in this population. Our study utilizes an effective rank-based FS approach to identify relevant proteomic target names in cross-validation for the classification. There is a bias–variance and time tradeoff related to cross-validation methods. Since we have a small number of cases with high-dimensional features, a five-fold cross-validation technique has been applied to mitigate bias. By leveraging the advantages of both filter and embedded FS methods, we obtained highly efficient results despite high-dimensional features and an NP-hard problem. The most appropriate method can vary by data type and dataset size depending on the problem considered due to the fact that there is no optimal method for all situations (e.g., no free lunch theorem). Additionally, the approach described here allowed for the identification of potentially important biomarkers needed to define CRT in proteomic panels and may be helpful for diagnostic or predictive operations meriting further study. Our proposed methodology has transferability to other numerical datasets, and it can be employed as a main algorithm for various tasks, similar problems, and different domains such as classification, pattern recognition, machine learning, and data mining. Future directions of this study include experimentation with other FS methods or their combinations and augmentation of these proteomic attributes with additional clinical, omics, radiation dosimetry, and imaging features to improve the performance results of the models towards the goal of personalizing management to improve patient outcomes. Alternative cross-validation methods also represent the future directions of this study. Further validation and FS methods specifically aimed at outcome measures, as they link to clinical data, are needed as progress is being made towards clinical utility in the detection, diagnosis, and treatment of glioblastoma.

## Figures and Tables

**Figure 1 cancers-15-02672-f001:**
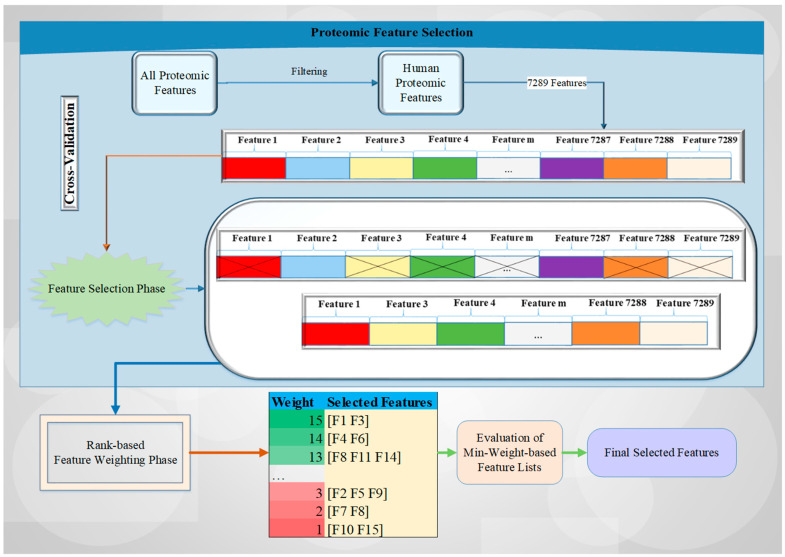
The brief overview of the proposed architecture.

**Figure 2 cancers-15-02672-f002:**
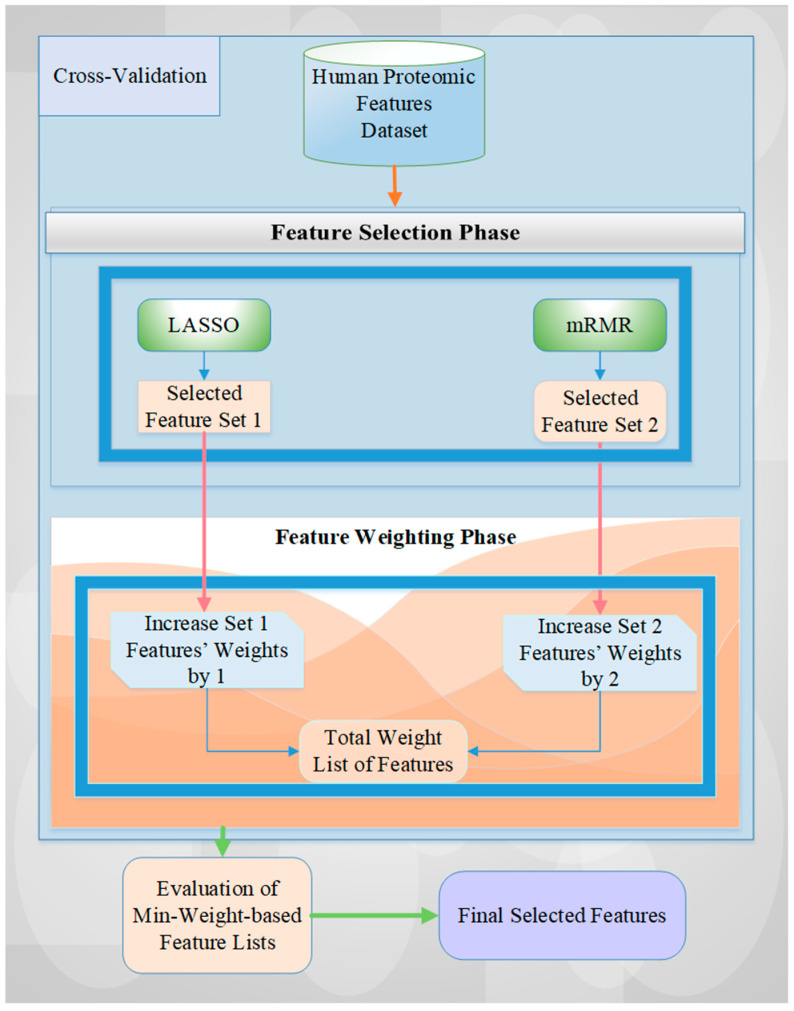
A detailed overview of the proposed methodology.

**Figure 3 cancers-15-02672-f003:**
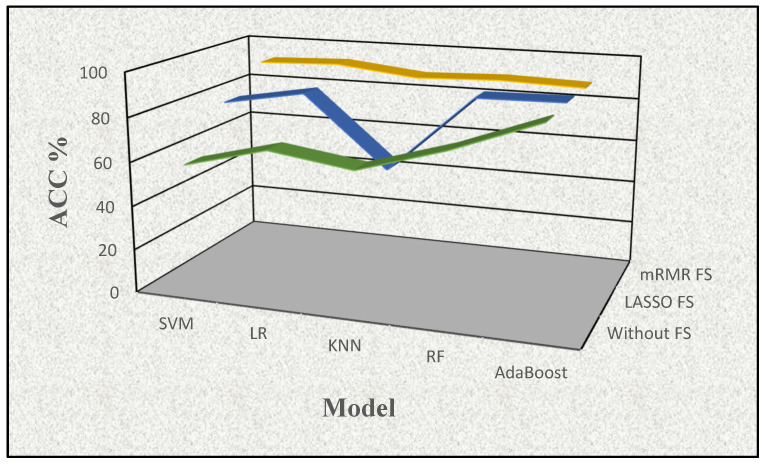
The visualization of the effects of the feature selection procedures with accuracy (ACC%) determined by a supervised learning method in conjunction with the feature selection approach (mRMR FS (yellow), LASSO FS (blue), and no FS (green)).

**Figure 4 cancers-15-02672-f004:**
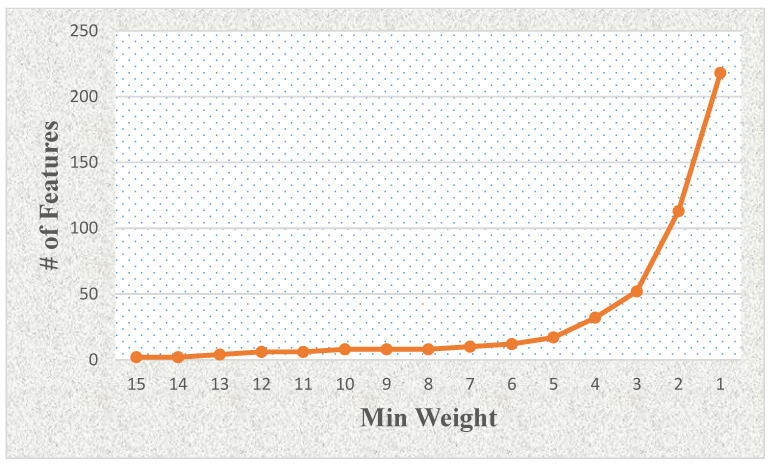
The effects of the number of features related to the minimum weight value using LASSO and mRMR-based feature selection with weighting methods.

**Figure 5 cancers-15-02672-f005:**
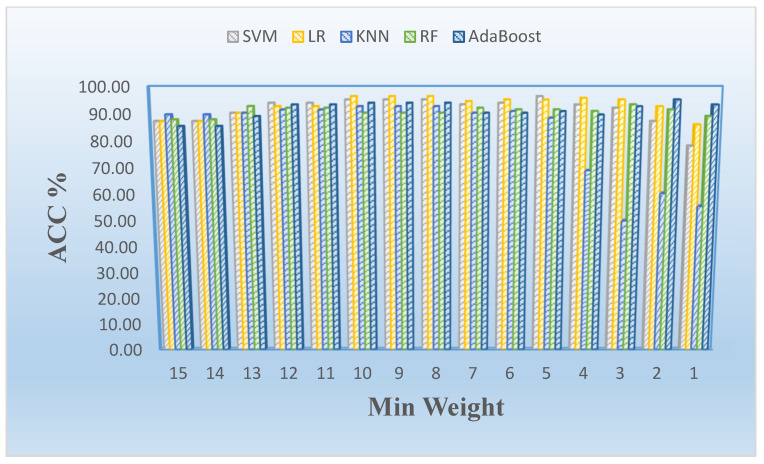
Mean accuracy rate (ACC) vs. minimum weight stratified by model employed in analysis.

**Figure 6 cancers-15-02672-f006:**
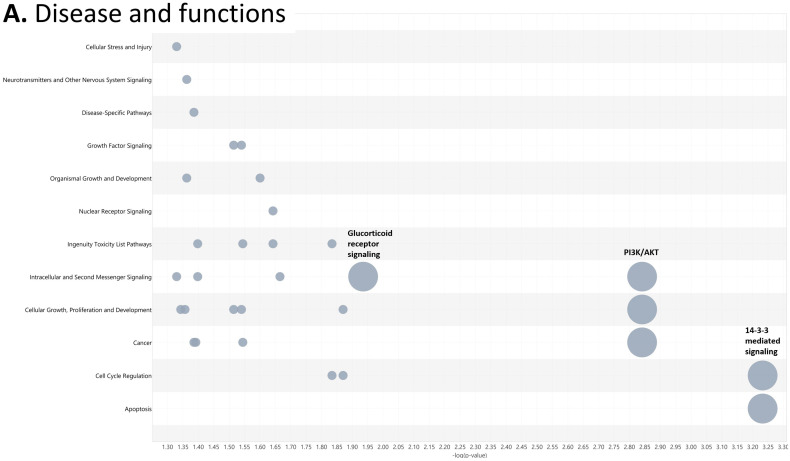
Ingenuity pathway analysis (IPA) employing the 8 ML-identified protein features for (**A**) disease and functions, (**B**) top canonical pathways, (**C**) role in PI3K/AKT signaling, (**D**) role in 14-3-3 mediated signaling.

**Figure 7 cancers-15-02672-f007:**
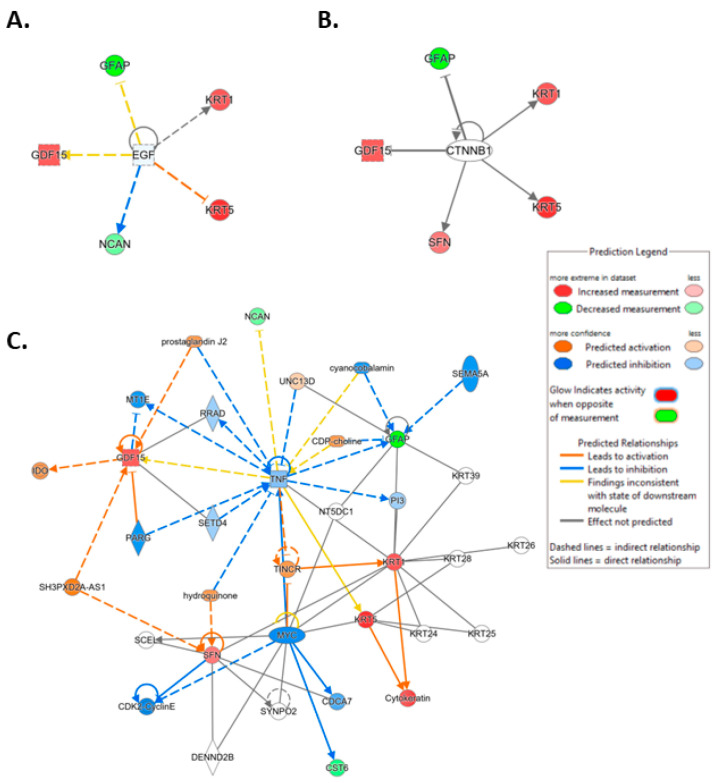
Ingenuity pathway analysis (IPA) carried out on April 5, 2023, illustrating linkage of the identified protein features to the top 2 upstream mediators (Appendix A). (**A**) Epidermal growth factor (EGF) (*p*-value of overlap 2.53 × 10^−7^). (**B**) Catenin beta 1 (CTNNB1) (*p*-value of overlap 2.41 × 10^−6^). (**C**) IPA-generated merged network for the 8 ML-identified proteins using the disease classification brain cancer.

**Table 1 cancers-15-02672-t001:** Accuracy rates: Five supervised learning models with or without feature selection. Color changes from red to green display performance results from the lowest (red) to the highest values (green).

ML-ACC	Without FS	LASSO FS	mRMR FS
**SVM**	57.860	78.674	91.515
**LR**	67.633	85.341	92.708
**KNN**	62.197	53.068	88.466
**RF**	73.826	88.466	89.659
**AdaBoost**	88.409	89.072	88.447

**Table 2 cancers-15-02672-t002:** Performance results (i.e., ACC%) using only LASSO-based feature selection and weighting methods. Color changes from red to green display performance results from the lowest (red) to the highest values (green). The bold value indicates the best result.

k	# of Features	SVM	LR	KNN	RF	AdaBoost
**5**	11	93.314	92.083	82.386	91.496	91.477
**4**	26	89.640	93.314	62.197	93.939	90.890
**3**	44	89.053	96.363	46.345	92.121	93.939
**2**	90	85.985	92.064	60.379	91.496	93.901
**1**	197	76.269	85.966	54.868	87.841	87.822

**Table 3 cancers-15-02672-t003:** Performance results (i.e., ACC%) using only mRMR-based feature selection and weighting methods. Color changes from red to green display performance results from the lowest (red) to the highest (green) values. The bold value indicates the best result.

k	# of Features	SVM	LR	KNN	RF	AdaBoost
**5**	5	86.004	88.428	87.235	87.841	87.197
**4**	7	90.890	92.708	90.265	91.496	91.496
**3**	8	95.152	96.364	92.708	90.871	93.920
**2**	11	92.708	96.364	92.689	90.284	94.508
**1**	34	92.102	92.102	87.254	92.121	90.871

**Table 4 cancers-15-02672-t004:** Mean performance results (i.e., ACC %, CV = 5) determined using both LASSO and mRMR-based feature selection with weighting methods. Color changes from red to green display performance results from the lowest (red) to the highest values (green). The bold value indicates the best result.

k	# of Features	SVM	LR	KNN	RF	AdaBoost
**15**	2	87.216	87.216	89.659	87.841	85.379
**14**	2	87.216	87.216	89.659	87.841	85.379
**13**	4	90.265	90.265	90.284	92.727	89.034
**12**	6	93.920	92.708	91.477	92.102	93.314
**11**	6	93.920	92.708	91.477	92.102	93.314
**10**	8	95.152	96.364	92.708	90.265	93.920
**9**	8	95.152	96.364	92.708	90.265	93.920
**8**	8	95.152	96.364	92.708	90.265	93.920
**7**	10	93.333	94.546	90.284	92.102	90.284
**6**	12	93.939	95.152	90.909	91.496	90.246
**5**	17	96.345	95.114	88.447	91.496	90.871
**4**	32	93.295	95.739	68.921	90.890	89.640
**3**	52	92.121	95.151	49.962	93.333	92.670
**2**	113	87.216	92.670	60.379	91.496	95.152
**1**	218	78.087	85.966	55.492	89.053	93.314

**Table 5 cancers-15-02672-t005:** The standard deviation of performance results (i.e., ACC %, CV = 5) determined using both LASSO and mRMR-based feature selection with weighting methods. Color changes from red to green display performance results from the lowest (red) to the highest values (green).

k	# of Features	SVM	LR	KNN	RF	AdaBoost
**15**	2	5.175	4.408	4.072	4.232	2.191
**14**	2	5.175	4.408	4.072	4.232	2.191
**13**	4	4.423	4.821	4.425	4.924	2.384
**12**	6	5.060	5.269	3.509	4.088	3.515
**11**	6	5.060	5.269	3.509	4.088	3.515
**10**	8	3.636	4.454	5.269	3.496	4.272
**9**	8	3.636	4.454	5.269	3.496	4.272
**8**	8	3.636	4.454	5.269	3.496	4.272
**7**	10	4.848	5.555	4.425	4.088	4.425
**6**	12	4.285	3.636	6.357	3.995	3.526
**5**	17	2.966	2.444	3.476	4.827	5.400
**4**	32	6.177	4.105	6.384	4.259	1.442
**3**	52	4.535	2.424	7.329	4.020	2.469
**2**	113	4.807	1.557	4.954	4.430	4.924
**1**	218	4.720	3.133	5.559	3.031	3.515

**Table 6 cancers-15-02672-t006:** Performance results without employing feature selection and feature weighting. Color changes from red to green display performance results from the lowest (red) to the highest values (green).

ML	ACC%	AUC	F1	PRE	REC	SPEC
**SVM**	57.860	0.415	0.518	0.698	0.515	0.690
**LR**	67.633	0.755	0.676	0.681	0.681	0.673
**KNN**	62.197	0.647	0.581	0.662	0.527	0.722
**RF**	73.826	0.808	0.744	0.768	0.746	0.737
**AdaBoost**	88.409	0.951	0.886	0.882	0.893	0.873

**Table 7 cancers-15-02672-t007:** Performance results employing LASSO and mRMR-based feature selection with weighting operation. Color changes from red to green display performance results from the lowest (red) to the highest values (green).

ML	ACC%	AUC	F1	PRE	REC	SPEC
**SVM**	95.152	0.989	0.949	0.975	0.928	0.976
**LR**	96.364	0.987	0.964	0.963	0.965	0.965
**KNN**	92.708	0.965	0.930	0.929	0.932	0.923
**RF**	90.265	0.978	0.902	0.885	0.928	0.876
**AdaBoost**	93.920	0.979	0.941	0.941	0.942	0.935

The best ACC% is 96.364, which was obtained with the Logistic Regression Model, and the minimum weight is 10. Selected Number of Features: 8. Best Feature (Biomarker) Set is as follows: ‘K2C5’, ‘MIC-1’, ‘CSPG3’, ‘GFAP’, ‘Proteinase-3’, ‘STRATIFIN’, ‘Cystatin M’, and ‘Keratin-1’ [59].

## Data Availability

Supporting data is available as supplementary material as listed above.

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
