# Peer review of "RadWise: A Rank-Based Hybrid Feature Weighting and Selection Method for Proteomic Categorization of Chemoirradiation in Patients with Glioblastoma"

_cancers, 2023, doi:10.3390/cancers15102672_

Round 1

Reviewer 1 Report

This manuscript describes a method where the most applicable proteomic features are used in order to analyse the sera of patients affected by glioblastomas undergoing chemoirradiation therapy. The authors present Radwise, a novel ranked-based feature weigthing method to identify all the most relevant proteomic parameters. 

The manuscript is well written and the conclusion raised by the authors  fully satisfy the data shown in the work. 

Author Response

Comment: This manuscript describes a method where the most applicable proteomic features are used in order to analyse the sera of patients affected by glioblastomas undergoing chemoirradiation therapy. The authors present Radwise, a novel ranked-based feature weighting method to identify all the most relevant proteomic parameters.

The manuscript is well written and the conclusion raised by the authors fully satisfies the data shown in work.

Response:  The authors appreciate the positive feedback from the reviewer. Thank you!

Reviewer 2 Report

I would like to recommend this manuscript for publication after minor revision:

1. Please remove the review mode from the article, which makes it look confusingï¼›

2. Figure 1,Figure 6 and Figure 7 can't be seen clearly, please provide the images with higher resolution.

3. How many times are repeated of the data in Figure 4 and Figure 5? Please expressed them as mean ± SD, n=?

4. Please provide the full name before using abbreviations for the first time.

Author Response

Please see attached PDF file. 

Thank you kindly for the review.

Reviewer 3 Report

In this study, the authors analyzed the proteomic data collected from patients with GBM pre and post-completion of concurrent chemoirradiation and found the most discriminative proteomic alternation that resulted from chemoirradiation. Finally, they presented a rank-based feature weighting method (RadWise) for proteomic categorization of chemoirradiation using LASSO and mRMR methods. This study is well presented and organized, the current revised manuscript is suitable for publication. 

Author Response

Comment : In this study, the authors analyzed the proteomic data collected from patients with GBM pre and post-completion of concurrent chemoirradiation and found the most discriminative proteomic alternation that resulted from chemoirradiation. Finally, they presented a rank-based feature weighting method (RadWise) for proteomic categorization of chemoirradiation using LASSO and mRMR methods. This study is well presented and organized, the current revised manuscript is suitable for publication.

Response :  The authors appreciate the positive feedback from the reviewer. Thank you!

Additional Explanations

A Grammarly check has been carried out and corrections applied.